# Maximizing Benefits under Harm Constraints: A Generalized Linear Contextual Bandit Approach

## Abstract

In many contextual sequential decision-making scenarios, such as dose-finding clinical trials for new drugs or personalized news article recommendation systems in social media, each action can simultaneously carry both benefits and potential harm. This could manifest as efficacy versus side effects in clinical trials, or increased user engagement versus the risk of radicalization and psychological distress in news recommendation. These multifaceted situations can be modeled using the multi-armed bandit (MAB) framework. Given the intricate balance of positive and negative outcomes in these contexts, there is a compelling need to develop methods which can maximize benefits while limiting harm within the MAB framework. This paper aims to address this gap. The primary contributions of this paper are two-fold: (i) We propose a novel contextual MAB model with the objective of optimizing reward potential while maintaining certain harm constraints. In this model both rewards and harm are governed by a generalized linear model with coefficients that vary based on the contextual variables. This flexibility allows the model to be broadly applicable for a wide range of scenarios. (ii) Building on our proposed generalized linear contextual MAB model, we develop an $\epsilon_t$-greedy-based policy. This policy is designed to strike an effective balance between the dual objectives of exploration-exploitation to achieve the desired trade-off between benefit and harm. We demonstrate that this policy achieves a sublinear $\mathcal{O}(\sqrt{T \log T})$ regret. Extensive experimental results are presented to support our theoretical analyses and validate the effectiveness of our proposed model and policy.

## 1 Introduction

The multi-armed bandit (MAB) problem is a classic framework in reinforcement learning, in which an agent makes sequential decisions to choose from multiple "arms" in each round, with each arm providing a stochastic reward. The objective is to find a strategy that maximizes the cumulative reward over a sequence of decisions. The fundamental issue is how to balance exploration (trying different arms to gain information) and exploitation (using the best option to maximize the reward). Various algorithms have been proposed to solve the MAB problem, such as $\epsilon_t$-greedy, upper confidence bound (UCB), and Thompson sampling, each offering different balances between exploration and exploitation. See for example Lattimore & Szepesvári (2020) and Slivkins (2019) for a thorough review.

The contextual multi-armed bandit (CMAB) problem extends the MAB framework by allowing each arm's expected reward to depend on some contextual variables. The CMAB problem has applications in numerous domains such as recommendation systems, personalized web services, and clinical trials. For example, within a news recommendation system, the articles to recommend (the arms) aim for the reward of a high click-through rate, which hinges on user-specific factors (the context) such as past clicking behavior. Similarly, in a clinical trial setting, the drug dosage (the arms) yields the reward of prolonged survival time, which depends on the patient-specific contextual information such as age, gender, and overall health status. See Lee et al. (2020) and Li et al. (2010).

In many practical scenarios, opting for a specific arm can result in both rewards and potential harm, and a focus on maximizing rewards alone might lead to unfavorable outcomes. For instance, within a

news recommendation system, promoting content similar to those that have previously elicited clicks from a particular user might yield a high click-through rate. However, this could also cultivate an echo chamber, causing emotional and psychological distress, polarize society, and even radicalize the user base. (see Comstock & Platania (2017); Houston et al. (2018)). Similarly, in a clinical trial context, a higher dosage may produce a positive outcome for the disease being treated (such as reducing tumor size), but it could also introduce more side effects, potentially harming the patient's overall health. Many efforts have been made to adopt MAB design in dose-finding clinical trials. See Aziz et al. (2021) and Villar et al. (2015) for example.

To tackle these challenging and intricate practical issues, we need to explore the CMAB problem jointly considering both rewards and harm at the same time. In the MAB literature, there have been discussions around the related topic of the multi-objective multi-armed Bandit (MOMAB), where the agent is assumed to have multiple objectives and a reward vector is revealed after each arm pull (see, for example, Drugan & Nowe (2013)). However, in recent MOMAB literature, the most popular approach, namely Pareto optimal front, cannot provide a safety guarantee. The safety-constrained MAB problems, on the other hand, poses some unrealistic assumptions. This motivates us to develop a framework to handle contextual multi-armed bandit problems with two opposing objectives, thus filling a gap in the literature. Surprisingly, by merely introducing a second objective, the possibility to reduce the problem to linear and apply LinUCB as in Li et al. (2017) for single objective has been ruled out. We specifically model the mean reward and harm of each arm using two separate varying coefficient generalized linear models, with the coefficients being dependent on the contextual variables. This framework is flexible enough to manage both discrete and continuous arms. We employ the Maximum Likelihood method for parameter estimation and develop an $\epsilon_t$-greedy algorithm with a harm function to address the exploration-exploitation trade-off. We prove the consistency of our parameter estimates and demonstrate that our $\epsilon_t$-greedy algorithm can achieve optimal regret. An extensive simulation study has been conducted which shows the superior performance of our proposed approach.

The rest of the paper is organized as follows. In Section 2, we provide a quick overview on related work, thus putting our work into comparative perspectives. In Section 3, we introduce the system model and problem formulation. Section 4 presents the our MAB policy design, which is followed by its main theoretical results. Section 5 demonstrates our numerical experiments and Section 6 concludes this paper.

## 2 RELATED WORK

In this section, we provide a quick overview on several related areas in the literature: 1) generalized linear bandit problems, 2) multi-objective bandits, 3) varying coefficient models, and 4) safety-constrained bandit problems, thus highlighting the differences and contributions of our work.

**1) Generalized linear bandit problems:** Generalized linear bandit problems with a single objective have been studied by Filippi et al. (2010), Li et al. (2017) and Kveton et al. (2020), who showed the advantage of generalized linear bandits over linear contextual bandits. Specifically, Filippi et al. (2010) studied stochastic generalized linear bandit and proposed algorithm called GLM-UCB, which achieves a regret of $\tilde{\mathcal{O}}(\sqrt{T})$ after $T$ rounds. Li et al. (2017) further considered contextual generalized linear bandit and demonstrated an $\tilde{\mathcal{O}}(\sqrt{T})$ regret for their UCB-GLM algorithm as well as $\tilde{\mathcal{O}}(\sqrt{T \log K})$ regret for SupCB-GLM algorithm, following the idea of Auer (2002) to create independent samples.

**2) Multi-objective bandits:** Drugan & Nowe (2013) introduced a stochastic multi objective multi-armed bandit (MOMAB) framework. UCB algorithms for MOMAB under both scalarized regret and Pareto regret have been proposed. Furthermore, Lu et al. (2019) considered multi-objective generalized linear bandit problems, where each arm possesses a feature that serves as the independent variable in the generalized linear model. This concept aligns with the approach in Aziz et al. (2021), who modeled toxicity and efficacy as a function of dose level in dosage finding clinical trials.

To highlight the difference in constructing context $x$ and feature of the arm $u$, a brief summary of the literature in CMAB and MOMAB is given in Table 1. As shown in Table 1, Filippi et al. (2010); Li et al. (2017) used another notion of context $x_{t,k}$ that is varying not only in rounds but also across arms. This notion is general but may impose difficulty on the practical meaning of the context. For

Table 1: Different formulations: $t$ represents rounds, $k$ represents arms, $x$ represents context, $u$ represents feature of the arms and $\eta$ represents parameter. $\mu(\cdot)$ is a generic notation for mean function.

| Algorithm | Mean Fuction | Context Changes w/ $t$ | Per-arm Context | Feature for Arms |
|---|---|---|---|---|
| 9; 18 | $\mu(x_{t,k}, \eta)$ | ✓ | ✓ | ✗ |
| 10 | $\mu(x_t, \eta_k)$ | ✓ | ✗ | ✗ |
| 3; 20 | $\mu(u_k, \eta)$ | - | - | ✓ |
| **Our work** | $\mu(x_t, u_k, \eta)$ | ✓ | ✗ | ✓ |

example, at a certain round $t$, patient information $x_t$ should not change when they are assigned different treatments (arms).

Another observation is the direct use of Pareto optimality, which is the current focus of MOMAB, does not provide a safety guarantee. An arm with both high harm and high reward may fall in the Pareto optimal front. However, a high harm may be unacceptable.

**3) Varying coefficient models:** Hastie & Tibshirani (1993) introduced varying coefficient models, which empower flexible estimation of regression models by allowing the coefficients to vary as smooth functions of other auxiliary variables. Varying coefficient models encompass a broad range of models, including generalized linear models, dynamic generalized linear models, and generalized additive models. Varying coefficient models are adopted in this paper to include a wide range of models rather than to propose a specific formulation.

**4) Safety-constrained bandit problems:** Another line of research that addresses the safety concerns in bandit problems can be found, for example, in (Amani et al., 2020). However, rather than observing both reward and harm as in this paper, only the reward is observed in (Amani et al., 2020). With only one set of responses available, parameters in safety constraints are assumed to be a known linear transformation of the parameters in the reward model. On a related note, Kazerouni et al. (2017) considered a conservative bandit setup with a known safe policy to serve as the baseline without modeling harm explicitly. Rather, the estimated optimal arm is pulled when it outperforms the baseline policy by a certain degree.

To the best of our knowledge, this paper is the first to apply varying coefficient models to GLM bandit problems. We show an $\mathcal{O}(\sqrt{T \log T})$ regret upper bound for a scalarized regret. This is the first regret bound for $\epsilon_t$-greedy algorithm in multi-objective generalized linear bandit. This result matches the state-of-the-art single-objective regret bound $\mathcal{O}(\sqrt{T \log T})$ for UCB-GLM algorithm presented in the work of Li et al. (2017).

## 3  SYSTEM MODEL AND PROBLEM FORMULATION

Consider a stochastic bandit setting with $K$ arms, which could represent dosage level in clinical trials or the level of controversial news recommendation amount. Suppose the $K$ arms form a discrete sampling from a continuous domain. Therefore, each arm is associated with a feature $u_k$ that is constant through the experiment. Without loss of generality, we can let $\underline{u} = (u_1, \cdots, u_K)$ with $u_1 \leq u_2 \leq \cdots \leq u_K$ to represent the $K$ arms. For example, in news recommendation systems, $\underline{u}$ could be the proportion of controversial content recommended to users. In clinical trials for new drugs, $\underline{u}$ may be the effective toxicity or efficacy assigned by physicians.

In each time step $t \in \{1, \cdots, T\}$, the learner receives a context $X_t$, pulls an arm $A_t$ from $\{1, \cdots, K\}$ and observes two outcomes: a harm effect $Y_t$ and a reward $Z_t$. Depending on the context $X_t$, each arm has a harm distribution $\mathcal{P}_{k,t}$ and a reward distribution $\mathcal{Q}_{k,t}$. Suppose that $Y_t \sim \mathcal{P}_{A_t,t}$ and $Z_t \sim \mathcal{Q}_{A_t,t}$ independently. Let $p_{k,t}$ and $q_{k,t}$ be the expectations of $\mathcal{P}_{k,t}$ and $\mathcal{Q}_{k,t}$, respectively.

We assume the context, harm outcome and reward are all bounded random variables with $\|X_t\| \leq r$ for some positive real number $r$, $Y_t \in [0, 1]$ and $Z_t \in [0, 1]$. Here $\|\cdot\|$ denotes the $\ell_2$ norm. For a positive definite matirx $A$, we let $\|x\|_A := \sqrt{x^T A x}$ denote the norm defined by $A$. The minimum and maximum eigen values of $A$ are written as $\lambda_{\min}(A)$ and $\lambda_{\max}(A)$, respectively. Finally, $1 : n := \{1, 2, \cdots, n\}$. Next, we define the unique features of our MAB model.

**1) Definition of the best arm:** Due to the existence of two metrics, in this paper, we define the best arm $k^*(t)$ at round $t$ as:

$$k^*(t) = \min\{\arg\max_k\{q_{k,t} - \lambda(p_{k,t} - \theta)_+\}\}, \tag{1}$$

where $\min$ is taken to break ties, $x_+$ is the positive part of $x$, $\theta$ is a pre-determined threshold for harm expectation, and $\lambda > 0$ is the penalty for choosing an arm with harm expectation larger than $\theta$. Both $\theta$ and $\lambda$ can be specified by practitioners based on their goals.

Therefore, a linear scalarization based on the penalty can be used to define the regret

$$R_T = \sum_{t=1}^{T}\{(q_t^* - q_{At,t}) + \lambda(p_{At,t} - \theta)_+ - \lambda(p_t^* - \theta)_+\}, \tag{2}$$

where $q_t^* = q_{k^*(t),t}$ is the expected rewards of the best arm $k^*(t)$, $p_t^* = p_{k^*(t),t}$ is the expected harm of arm $k^*(t)$.

It is also insightful to consider other alternatives of the best arm definition. For example, one may desire to define the best arm as

$$k'(t) = \min\{k : q_{k,t} = \max_{l:p_{l,t}\leq\theta} q_{l,t}\} \tag{3}$$

and choose $\hat{k}'(t) = \min\{k : \hat{q}_{k,t} = \max_{l:\hat{p}_{l,t}\leq\theta}\hat{q}_{l,t}\}$ during the trial. This approach is adopted by Lee et al. (2020) and Aziz et al. (2021) in the pure exploration problem. Note that the best arm definition in equation 1 is an attempt to solve equation 3 approximately. When $\lambda = \infty$, the best arm defined in equation 1 will be the same as equation 3. Therefore, If we are strictly conservative about the harm, the definition in equation 3 would be more desirable. The effect of $\lambda$-values on the chosen arm's harm probability is provided in the Appendix, showing equation 1 is a good approximation to equation 3 when $\lambda$ is large. At the same time, the penalty term in equation 1 allows a soft constraint on harm when we are not absolutely strict about it. In addition, if we take threshold $\theta = 0$, then equation 1 imposes penalty whenever there is harm effect, but equation 3 will give an empty set. Overall, we believe the best arm definition equation 1 provides more generality.

**Remark 3.1.** *Under equation 3, $k'(t)$ does not necessarily yield the largest $q_{k'(t),t} - (\lambda(p_{k'(t),t} - \theta)_+$ among all $\{q_{k,t} - (\lambda(p_{k,t} - \theta)_+\}_{k=1}^{K}$, so the regret equation 2 may admit negative terms and does not suit the best arm definition in equation 3.*

**2) Varying coefficient generalized linear models for harm and benefits:** A generalized linear model for the harm takes $u_k$ as the covariate, and it is common to assume that the expectation of harm increases with $u_k$. We consider the parameters in the GLM to vary with the context $x$. When the learner receiving context $x \in \mathbb{R}^d$, if arm $k$ is pulled, the expectation of harm $p$ is given by

$$p(x, k; \beta) = g(\zeta(x, k; \beta))), \tag{4}$$
$$\zeta(x, k; \beta) = b_0(x; \beta_0) + b_1(x; \beta_1)u_k, \tag{5}$$

where $g$ is the inverse of link function for harm model, $\zeta$ denotes the systematic component, $u_k$ is the feature associated with arm $k$, and $b_0(x; \beta_0)$ and $b_1(x; \beta_1)$ are functional coefficients. If $b_0(x; \beta_0) \equiv b_0$ and $b_1(x; \beta_1) \equiv b_1$ for some constants $b_0$ and $b_1$, it is the common generalized linear model.

For now, we assume parametric models $b_0(x) = \Phi(x)^T\beta_0$ and $b_1(x) = \Phi(x)^T\beta_1$ for some known transformation $\Phi(\cdot)$ from $\mathbb{R}^d$ to $\mathbb{R}^{d_1}$. It is interesting future work to extend $\Phi(\cdot)$ to a spline basis to allow more flexible models.

The link function $g^{-1}$ for generalized linear models is a monotone increasing function. Based on the distribution of the response, different link functions can be used. For example, the canonical link function for Gaussian distributed response is the identity link $g^{-1}(x) = x$, and the canonical link function for Bernoulli response is the logit link $g^{-1}(x) = \log\frac{x}{1-x}$.

The conditional distribution of harm $Y_t$ given context $X_t$ and arm $A_t$ is from the exponential family in classic generalized linear models. The conditional density of $Y_t|(X_t = x, A_t = k)$ can be written as

$$f(y|x, k) = \exp\{\phi[y\zeta(x, k; \beta) - m_y(\zeta(x, k; \beta))] + c(y, \phi)\},$$

where $\phi$ is the known dispersion parameter, $m_y(\zeta)$ satisfies $m'_y(\zeta(x,k;\beta)) = p(x,k;\beta) = \mathbb{E}(Y_t|X_t = x, A_t = k)$, and $c(y,\phi)$ is the normalization function that does not involve $\zeta$.

Conditioning on $\{X_1, A_1, X_2, A_2, \cdots, X_t, A_t\}$, $Y_1, Y_2, \cdots, Y_t$ are independent. The log-likelihood function of $\beta$ can be written as:

$$\ell_t(\beta) = \sum_{s=1}^{t} \left\{ \phi[Y_s\zeta(X_s, A_s; \beta) - m_y(\zeta(X_s, A_s; \beta))] + c(Y_s, \phi) \right\}$$

$$= \phi\sum_{s=1}^{t}[Y_s\zeta(X_s, A_s; \beta) - m_y(\zeta(X_s, A_s; \beta))] + \text{constant}.$$

As a result, the MLE of $\beta$ can be defined as

$$\hat{\beta}_t = \arg\max_{\beta} \sum_{s=1}^{t}[Y_s\zeta(X_s, A_s; \beta) - m_y(\zeta(X_s, A_s; \beta))]. \tag{6}$$

In this paper, we let

$$Y_t = p(X_t, A_t; \beta) + e_t^{(y)},$$

where $\{e_t^{(y)}\}_{t=1}^{T}$ are independent zero-mean sub-Gaussian noise that is independent of $X_t$ and $A_t$. Note that if $Y_t|(X_t, A_t)$ follows Gaussian, Bernoulli or any bounded distribution, $e_t^{(y)}$ can be shown to be sub-Gaussian.

Similar to the harm model, we assume for expected reward $q$,

$$q(x, k; \gamma) = h(\xi(x, k; \gamma)), \tag{7}$$
$$\xi(x, k; \gamma) = c_0(x; \gamma_0) + c_1(x; \gamma_1)u_k + c_2(x; \gamma_2)u_k^2, \tag{8}$$

where $h$ is the inverse of the link function for reward model, $\xi$ denotes the systematic component, $c_0(x), c_1(x)$ and $c_2(x)$ are functional coefficients. We use a quadratic model to allow non-monotonic changes as $k$ increases. An increasing-plateau model may also be considered, see more possible models in Pinheiro et al. (2014) for example. Here we also use parametric models $c_0(x) = \Psi(x)^T\gamma_0$, $c_1(x) = \Psi(x)^T\gamma_1$ and $c_2(x) = \Psi(x)^T\gamma_2$ with a known transformation $\Psi(\cdot) : \mathbb{R}^d \to \mathbb{R}^{d_2}$.

Similarly, the MLE for $\gamma$ can be expressed as

$$\hat{\gamma}_t = \arg\max_{\gamma} \sum_{s=1}^{t}[Z_s\xi(X_s, A_s; \gamma) - m_z(\xi(X_s, A_s; \gamma))], \tag{9}$$

for $m'_z(\xi(x, k; \gamma)) = q(x, k; \gamma)$. The reward model can be written as $Z_t = q(X_t, A_t; \gamma) + e_t^{(z)}$, where $\{e_t^{(z)}\}_{t=1}^{T}$ are independent zero-mean sub-Gaussian noise.

Let $\beta = (\beta_0^T, \beta_1^T)^T$, $\gamma = (\gamma_0^T, \gamma_1^T, \gamma_2^T)^T$, and $\eta = (\beta^T, \gamma^T)^T$ denote all the parameters of interest.

The varying coefficient models are a natural extension to Aziz et al. (2021) and Lu et al. (2019), who only define $u_k$ to characterize the arms rather than utilize information $X_t$ about the task of each round.

Except the boundedness of context, harm and reward, the following assumptions on the smoothness of link functions are needed in order to establish the bound on regret.

A1: $\kappa_y := \inf\limits_{\substack{\|x\|\leq r, \\ \|\beta'-\beta\|\leq 1}} g'(\zeta(x, k; \beta')) > 0$ and $\kappa_z := \inf\limits_{\substack{\|x\|\leq r, \\ \|\gamma'-\gamma\|\leq 1}} h'(\xi(x, k; \gamma')) > 0$.

A2: $g$ and $h$ are twice differentiable. The first and second derivatives of $g$ are bounded from above by $L_g$ and $M_g$, and the first and second derivatives of $h$ are bounded from above by $L_h$ and $M_h$, respectively.

Assumptions A1 and A2 have been used by Li et al. (2017). A1 controls the local behavior of $g'$ and $h'$ when using parameter value $\beta'$ and $\gamma'$ near the true parameter value $\beta$ and $\gamma$. A1 is necessary for the convergence of parameter estimates. It is easy to verify A1 and A2 hold for identity link and logit link. Specifically, for identity link, $L = 1$ and $M = 0$. For logit link $L = M = 1/4$.

Since we assume parametric models $\Phi(\cdot)$ and $\Psi(\cdot)$, the linear predictors for harm and reward can be written as

$$\zeta(x, k; \beta) = (\Phi(x)^T, u_k \Phi(x)^T)\beta,$$
$$\xi(x, k; \gamma) = (\Psi(x)^T, u_k \Psi(x)^T, u_k^2 \Psi(x)^T)\gamma.$$

Define $W_t = (\Phi(X_t)^T, u_{A_t}\Phi(X_t)^T)^T \in \mathbb{R}^{2d_1}$ and $V_t = (\Psi(X_t)^T, u_{A_t}\Psi(X_t)^T, u_{A_t}^2\Psi(X_t)^T)^T \in \mathbb{R}^{3d_2}$, the design matrices for harm and reward model up to round $t$ can be represented as $(W_1, W_2, \cdots, W_t)^T$ and $(V_1, V_2, \cdots, V_t)^T$.

The boundedness of $\|W_t\|$ and $\|V_t\|$ can be verified as long as the transformations $\Phi(\cdot)$ and $\Psi(\cdot)$ are continuous, and $\max_{1 \leq k \leq K} u_k < \infty$. Without loss of generality, we assume $W_t$ and $V_t$ are normalized:

A3: $\|W_t\| \leq 1$ and $\|V_t\| \leq 1$ for all $1 \leq t \leq T$.

A4: $\mathbb{E}[W_t W_t^T]$ and $\mathbb{E}[V_t V_t^T]$ are positive definite.

## 4 POLICY DESIGN AND MAIN RESULTS

In this paper, we propose an $\epsilon_t$-greedy-based algorithm for solving the problem. To obtain an initial estimate of $\eta$, each arm is pulled $m$ rounds at the beginning of the experiment. In clinical trials, however, the initialization and exploration should be more carefully designed to avoid exposing patients to high toxicity, see Aziz et al. (2021) for using an "admissible set" or "admissible doses" for example.

---

**Algorithm 1:** $\epsilon_t$-greedy Algorithm with harm penalty

---

**input:** Time horizon $T$, exploration rate $\epsilon_t \in (0, 1)$, penalty $\lambda \in (0, \infty)$, harm threshold $\theta \in (0, 1)$, initialization rounds $m$

**for** $t = 1, \cdots, m \times K$ **do**
    Sample from each arm $m$ times, record context $X_t$ and response $Y_t, Z_t$
**end**
**for** $t = m \times K + 1, \cdots, T$ **do**
    Obtain maximum likelihood estimate $\hat{\eta}_{t-1}$ based on $X_{1:t-1}, A_{1:t-1}, Y_{1:t-1}$ and $Z_{1:t-1}$ as in
      equation 6 and equation 9
    Identify best arm $\hat{k}_t = \arg\max_k q(X_t, k; \hat{\gamma}_{t-1}) - \lambda(p(X_t, k; \hat{\beta}_{t-1}) - \theta)_+$
    Sample from $\hat{k}_t$ w.p. $1 - \epsilon_t + \dfrac{\epsilon_t}{K}$,
    and sample from arms in $\{1, \cdots, K\}\backslash\{\hat{k}_t\}$ w.p. $\dfrac{\epsilon_t}{K}$
    Record context $X_t$, choice $A_t$ and responses $Y_t, Z_t$
**end**
**output:** Parameter estimates $\hat{\eta}_T$

---

Shrinking exploration probability $\epsilon_t = \min\left\{1, C\dfrac{\log t}{t}\right\}$ is taken as suggested by Cesa-Bianchi & Fischer (1998) with constant $C > 0$.

The single-objective setting in (Li et al., 2017) allows a straightforward application of LinUCB based on the linear systematic component. However, the two-objective setup in our paper already precludes the possibility to directly reduce the problem to a linear case. Specifically, in the single-objective case, since the link function $h(\xi)$ is monotonically increasing in $\xi$, as long as $\xi(x, k; \gamma) \geq \xi(x, j; \gamma)$ for arms $k, j$, we know $h(\xi(x, k; \gamma)) \geq h(\xi(x, j; \gamma))$. Thus, it suffices to find the arm $k$ that maximizes $\xi(x, k; \gamma)$ as well as design an upper confidence bound for $\xi(x_t, k; \gamma)$. However, our scalarized

reward in equation 2 is *no longer* monotone in $\xi$ or $\zeta$, so the problem cannot be decomposed into two linear pieces and dealt with separately.

To establish an upper bound for regret of Algorithm 1, it is essential to show convergence of parameter estimates. Recall Theorem 1 from Li et al. (2017) for convergence of parameter estimates. For simplicity, we state the theorem for $W_t$ and $Y_t$. Note that the result also holds for $V_t$ and $Z_t$.

**Lemma 1** (Theorem 1 in (Li et al., 2017)). *Define $G_t = \sum_{s=1}^{t} W_s W_s^T$, and let $\delta > 0$ be given. Furthermore, assume that*

$$\lambda_{\min}(G_t) \geq \frac{512 M_g^2 \sigma_y^2}{\kappa_y^4} \left( 4d_1^2 + \log \frac{1}{\delta} \right). \tag{10}$$

*Then, with probability at least $1 - 3\delta$, the maximum likelihood estimator $\hat{\beta}_t$ satisfies, for any $w \in \mathbb{R}^{2d_1}$,*

$$|w^T (\hat{\beta}_t - \beta)| \leq \frac{\sigma_y}{\kappa_y} \sqrt{\log(1/\delta)} \|w\|_{G_t^{-1}}. \tag{11}$$

Condition equation 10 is satisfied when there is a sufficient number of independent samples collected in a trial of the multi-armed bandit problem. The sample size needed for equation 10 to hold is also stated in Li et al. (2017) and is proficed in the Appendix in order for our work to be self-contained.

Based on the statistical property of parameter estimates, the theorem below shows an $\mathcal{O}(\sqrt{T \log T})$ regret bound.

**Theorem 1.** *Let $\tau = m \times K$ denote the number of initialization rounds. If we run the Algorithm 1 with*

$$\tau = \max \left\{ \left( \frac{C_1 \sqrt{2d_1} + C_2 \sqrt{\log(2/\delta)}}{\lambda_{\min}(\Sigma_1)} \right)^2 + \frac{2B_1}{\lambda_{\min}(\Sigma_1)}, \right.$$

$$\left. \left( \frac{C_1 \sqrt{3d_2} + C_2 \sqrt{\log(2/\delta)}}{\lambda_{\min}(\Sigma_2)} \right)^2 + \frac{2B_2}{\lambda_{\min}(\Sigma_2)} \right\},$$

*then with probability at least $1 - 3\delta$, the regret of the algorithm is upper bounded by*

$$R_T \leq (1 + \lambda)\tau \tag{I}$$

$$+ (1 + \lambda) \left( C \frac{(\log T)^2}{2} - C \frac{(\log \tau)^2}{2} + \sqrt{\frac{T - \tau}{2} \log \frac{1}{\delta}} \right) \tag{II}$$

$$+ 2\lambda L_g \frac{\sigma_y}{\kappa_y} \sqrt{4d_1(T - \tau) \log \frac{6}{\delta} \log \frac{T}{2d_1}} + 2L_h \frac{\sigma_z}{\kappa_z} \sqrt{6d_2(T - \tau) \log \frac{6}{\delta} \log \frac{T}{3d_2}}. \tag{III}$$

*Here, we may choose*

$$B_1 = \max \left\{ 1, \frac{512 M_g^2 \sigma_y^2}{\kappa_y^4} \left( 4d_1^2 + \log \frac{6}{\delta} \right) \right\},$$

$$B_2 = \max \left\{ 1, \frac{512 M_h^2 \sigma_z^2}{\kappa_z^4} \left( 9d_2^2 + \log \frac{6}{\delta} \right) \right\},$$

*and the second moments $\Sigma_1 := \mathbb{E}[W_t W_t^T]$ and $\Sigma_2 := \mathbb{E}[V_t V_t^T]$ exist following Assumption A3.*

*Sketch of proof.* The regret $R_T$ can be decomposed into three parts: regret from exploration round 1 to $\tau$ (I), regret from exploration rounds (II) and regret from greedy rounds (III).

For the regret at time step $t$, $q_t^* - q_{A_t,t} + \lambda(p_{A_t,t} - \theta)_+ - \lambda(p_t^* - \theta)_+ \leq 1 + \lambda$ always holds since $p_{k,t}, q_{k,t} \in [0, 1]$. Therefore, the sum of regret from round 1 to $\tau$ is bounded by $(1 + \lambda)\tau$.

The probability of exploration $\epsilon_t = C \frac{\log t}{t}$ is shrinking with round $t$. Thus, by the Hoeffding's inequality, the number of exploration rounds is bounded from above by $C \frac{(\log T)^2}{2} - C \frac{(\log \tau)^2}{2} + \sqrt{\frac{T - \tau}{2} \log \frac{1}{\delta}}$ w.p. at least $1 - \delta$, resulting in term (II) in the regret bound.

If the algorithm does not explore at round $t$, due to the greedy characteristic, we have

$$h(V_t^T \hat{\gamma}_{t-1}) - \lambda(g(W_t^T \hat{\beta}_{t-1}) - \theta)_+ \geq h(V_{t*}^T \hat{\gamma}_{t-1}) - \lambda(g(W_{t*}^T \hat{\beta}_{t-1}) - \theta)_+,$$

which implies

$$
\begin{aligned}
&h(V_{t*}^T \gamma) - h(V_t^T \gamma) + \lambda(g(W_t^T \beta) - \theta)_+ - \lambda(g(W_{t*}^T \beta) - \theta)_+ \\
\leq &h(V_{t*}^T \gamma) - h(V_t^T \gamma) + \lambda(g(W_t^T \beta) - \theta)_+ - \lambda(g(W_{t*}^T \beta) - \theta)_+ \\
&+ h(V_t^T \hat{\gamma}_{t-1}) - \lambda(g(W_t^T \hat{\beta}_{t-1}) - \theta)_+ - h(V_{t*}^T \hat{\gamma}_{t-1}) + \lambda(g(W_{t*}^T \hat{\beta}_{t-1}) - \theta)_+ \\
\leq &L_h |V_{t*}^T (\hat{\gamma}_{t-1} - \gamma)| + L_h |V_t^T (\hat{\gamma}_{t-1} - \gamma)| + \lambda L_g |W_t^T (\hat{\beta}_{t-1} - \beta)| + \lambda L_g |W_{t*}^T (\hat{\beta}_{t-1} - \beta)|,
\end{aligned}
$$

and the last line can be bounded using Lemmas 1 and 4 to obtain the third part of the regret bound. Due to space limitation, the detailed proof is deferred to the Appendix.

## 5 NUMERICAL EXPERIMENTS

We test the varying coefficient model against four baseline models in 100 trials. In each trial, we run the algorithms for $T = 5,000$ rounds. $K = 7$ arms are used in the simulation with $\underline{u} = (0.1, 0.2, \cdots, 0.7)$. The penalty is chosen to be $\lambda = 1$, and harm threshold $\theta = 0.33$.

**1) Methods for comparison:** We consider four baseline methods, all of which under the $\epsilon_t$-greedy framework. The first baseline method ignores the harm effect, and only targets at maximizing the reward. The second baseline method ignores the context, which means the functional coefficients are all treated as constants.

The third one is to learn $K$ separate models independently for each of the $K$ arms. This method is used in Goldenshluger & Zeevi (2013). In this method, we assume arm $k$ has parameters $\beta^{(k)}$ and $\gamma^{(k)}$ with harm model $p(x, \beta^{(k)}) = g(x^T \beta^{(k)})$ and reward model $q(x, \gamma^{(k)}) = h(x^T \gamma^{(k)})$. When running the algorithm, only data from the history of arm $k$ is used to estimate $\hat{\beta}_t^{(k)}$ and $\hat{\gamma}_t^{(k)}$. Therefore, for arms $j \neq k$, the parameters $\beta^{(j)}, \gamma^{(j)}$ and $\beta^{(k)}, \gamma^{(k)}$ are learned separately.

The fourth baseline method under consideration is to bin the continuous context into $N$ categories $C_1, \cdots, C_N$, then learn a context-free model within each category. The binned context is considered by Perchet & Rigollet (2013) and Li et al. (2019). For $x$ belonging to category $C_i$, we assume there are parameters $\beta^{(i)} \in \mathbb{R}^2$ and $\gamma^{(i)} \in \mathbb{R}^3$, and model expected harm as $p(x, u_k; \beta^{(i)}) = (1, u_k)\beta^{(i)}$ as well as expected reward $q(x, u_k; \gamma^{(i)}) = (1, u_k, u_k^2)\gamma^{(i)}$. When context $X_t$ arrives, we first decide which category $X_t$ belongs to, say $C_i$, then use history data $\{1 \leq s \leq t - 1 : X_s \in C_i\}$ to estimate $\hat{\beta}_t^{(i)}$ and $\hat{\gamma}_t^{(i)}$. $N = 3$ categories are used in the simulation.

**2) Data generation:** We consider a simple example where $x \in \mathbb{R}^1$, and $\Phi(x) = \Psi(x) = (1, x)^T$ for the varying coefficient model. The context $X_t$ is simulated i.i.d. from a Uniform distribution on $[0, 1]$. The harm $Y_t$ and reward $Z_t$ are Bernoulli random variables with mean functions specified by equation 4 and equation 7, respectively. The inverse link functions $g$ and $h$ are both expit functions. The surfaces for the expectation of harm and reward are given in Figure 1.

We also run the oracle method, where the true $\beta$ and $\gamma$ values are known, to identify the best arm in each round. For the five methods under comparison, we report the regret of each method in Figure 2 (a). Figure 2 (b) gives the count $\sum_{s=1}^t I\{p_s > \theta\}$ for the chisen arm, reflecting the safety feature of each method. Due to space limit, more details on computation as well as additional results on different $\lambda$-value are deferred to the Appendix.

The advantage of varying coefficient models becomes evident when examining the results depicted in both plots. In Figure 2(a), the regret for varying coefficient models is constantly lower than the other four methods. In Figure 2(b), the varying coefficient model is also comparable with the oracle method on the safety side. These results emphasize the effectiveness of varying coefficient models in making more informed and advantageous decisions.

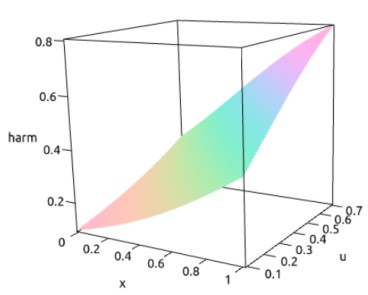 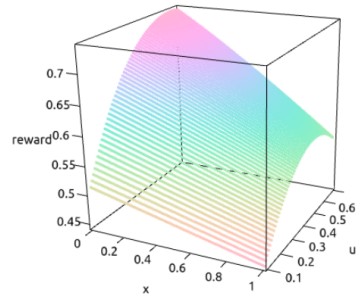

(a) Surface of expectation of harm on feature $u$ and context $x$.

(b) Surface of expectation of reward on feature $u$ and context $x$.

Figure 1: Note reward is not monotonely increasing with level ($u$), but harm is assumed to be monotone in level ($u$).

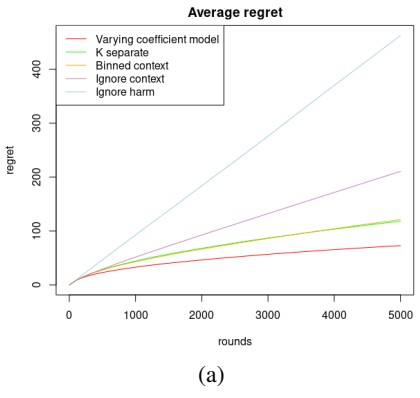 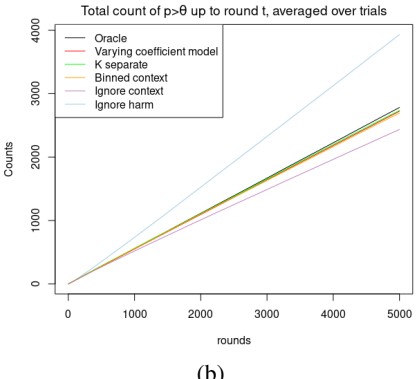

(a)

(b)

Figure 2: (a) The cumulative regret averaged across 100 trials. (b)To reflect the safety feature of each method, we calculate how many times the method chooses an arm with harm probability $p > \theta$ up to round $t$. Then the count is averaged over 100 trials.

# 6 CONCLUSION

In this paper, we considered a new contextual multi-armed bandit framework, where both benefits and harm are the consequence of pulling an arm. By including the penalty of harm in the regret formulation, we balanced between maximizing benefits and controlling harmful outcomes. Our proposed model used contextual variables to construct the varying coefficient generalized linear model, allowing flexible model specification. We proposed an $\epsilon_t$-greedy algorithm to make decisions sequentially. We established an $\mathcal{O}(\sqrt{T \log T})$ regret bound for the proposed $\epsilon_t$-greedy algorithm under varying coefficient generalized linear models. Compared to the baseline models, varying coefficient models have shown an advantage in both minimizing regret and controlling harm. The varying coefficient models successfully captured the connection between arms and utilized the information provided by contextual variables. To allow greater model flexibility, the functional coefficients may be estimated by nonparametric methods instead of parametric models. The statistical properties of such nonparametric estimations are worth investigating. Also, the MLE of $\eta$ becomes computationally expensive as the time horizton $T$ increases. Online update of parameters is worth considering for large $T$ to further promote the applicability of the proposed method. See Chen et al. (2021) for example. Another approach to tackle the balance of benefit and harm is the Pareto optimality, which may be adapted more carefully if safety guarantee is needed. Finally, in the current analysis of regret, the convergence of parameter estimates was solely based on the initialization rounds. However, $\epsilon_t$-greedy algorithm collects more independent samples during the exploration. A careful analysis to utilize the independent samples in exploration may lead to a better regret bound.

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
