# OpenReview forum: "Maximizing Benefits under Harm Constraints: A Generalized Linear Contextual Bandit Approach"
_ICLR.cc/2024/Conference — Submitted to ICLR 2024_

### Official Review · Reviewer_yL5v · 2023-10-30

**Soundness:** 3 good
**Presentation:** 2 fair
**Contribution:** 1 poor
**Rating:** 3
**Confidence:** 4

**Summary:**

The paper studies a sequential decision-making problem where the goal is to maximize the reward while containing a harm measure. They propose to model the reward and harm responses as a GLM and use a simple $\epsilon$-greedy algorithm to solve the problem.

**Strengths:**

1. The paper reads smoothly and is clear in all aspects.

**Weaknesses:**

1. Significance: the paper considers just an instance of constrained bandits. There are several related works similar to this setting (see Section 2). Even the modeling is a special case of algorithms 9,18 in Table 1 (consider concat$(x_i, u_k) \equiv x_{t,k}$). The $\epsilon$-greedy algorithm is a very basic algorithm for any bandit algorithm. Why not use more optimal algorithms like Thompson sampling or UCB?
2. The experimental results only include synthetic studies and very minimal. Experimenting with real-world public datasets could be insightful for instance for the model misspecification accounts.

**Questions:**

1. How can a practitioner come up with $\theta$ and $\lambda$? these are hyperparameters of your algorithm and do not directly translate into their goals.
2. Depending on the problem instance, $\kappa$ could be arbitrarily small. Then the regret bounds in Thm. 1 are meaningless. How can you address this issue?

---

> ### Author Response · Authors · 2023-11-22
>
> Thank you for your thoughtful comments! We address your concerns below.
> * **The paper considers just an instance of constrained bandits. Even the modeling is a special case of algorithms [9,18] in Table 1 (consider concat $(x_t, u_k) = x_{t,k})$.**
>
>   The safety-constrained bandit problems mentioned in part 4) of Section 2 are quite different from our approach. Amani et al (2020) assume we only observe one set of responses to the reward $Z_t$, and $Z_t = g(X_t^T\beta)+e_t$, where $g(\cdot)$ is the inverse of link function (known), $\beta$ is an unknown parameter and $e_t$ is noise. They use a safety constraint $h(BX_t^T\beta)\leq c$ with known function $h$ and matrix $B$. In practice, modeling harm based on reward as well as assuming a known linear transformation $B$ may be limited. The other method by Kazerouni et al(2017) assumed the existence of a known safe policy as a reference policy, which may also be limited.
>   Our modeling is *not* a special case of the algorithm in [9,18], even though we may concat $x_t$ and $u_k$. Since [9, 18] assume a linear component $x_{t,k}^T\beta$. If we simply concat $x_t$ and $u_k$ under the model of [9, 18], we are assuming an additive relation of $x_t$ and $u_k$: $x_{t,k}^T\beta = x_t^T\beta_1 + u_k^T\beta_2$ for $\beta = (\beta_1, \beta_2)$. Our model (see equation 4,5,7,8) is more general and considers interaction between $x_t$ and $u_k$.
>
>   Overall, our method cannot possibly solve all instances of constrained bandit problems but aims at fitting a gap in literature. The formulation to use varying coefficient generalized linear models also extends the modeling choices in literature.
> * **How can a practitioner come up with $\lambda$ and $\theta$?**
>   We do not consider $\theta$ as a hyperparameter. $\theta$ is the upper bound of harm that practitioners want to force in the experiment. A major example of this work is clinical trials where the typical $\theta$ value can be 0.166 or 0.33. $\lambda$ will affect the probability of the event "chosen arm has harm $p >\theta$". So intuitively, if practitioners want this probability to be low, they can specify an arbitrarily large $\lambda$.
>
> * **Depending on the problem instance, $\kappa$ could be arbitrarily small. Then the regret bounds in Thm. 1 are meaningless. How can you address this issue?**
>   $\kappa$ controls the rate of convergence of the maximum likelihood estimator in GLM. If $\kappa$ is small, the the parameter $\hat\beta_t$ and $\hat\gamma_t$ will converge to $\beta$ and $\gamma$ slowly. In this case, it's not surprising the regret bound will be large. If you make $\kappa$ arbitrarily small, then the problem at hand is arbitrarily difficult, thus the result is not ideal. We do not think this can be resolved by a particular bandit algorithm. So it is not fair to say it is a drawback of our work.
>
> * **No real data experiment.**
>   Since there is no real experiment conducted according to the algorithm, we cannot provide real data analysis.

---

### Official Review · Reviewer_4G5T · 2023-10-30

**Soundness:** 3 good
**Presentation:** 4 excellent
**Contribution:** 3 good
**Rating:** 6
**Confidence:** 3

**Summary:**

In this paper, the authors address the setting of Multi-Objective Contextual Multi Armed Bandits.
The motivation for the work is the necessity, in many practical scenarios, to execute actions considering the positive as well as the negative outcome that such actions entail. The main motivating example cited in the paper is that of clinical trials.
The setting addressed by the authors contains information on the context $X_t$, common to all arms and varying at each round, as well as features for each arm, constant across the rounds.
The authors propose a method to tackle such a problem by modelling the mean reward and harm of every arm via two varying coefficient GLMs.
The parameters for such models are to be estimated through MLE.
Finally, after an initial uniform exploration of the arms, an $\epsilon$-greedy algorithm is employed to address the exploration-expolitation trade-off.

**Strengths:**

The paper, as the authors also acknowledge, places itself in a field where many results have been proposed. Some of the existing results have many aspects in common with the authors' results, although there are some differences.
The paper is well written, the setting is clear and introduced clearly, the notation is clear, as are the results.
In my opinon, the biggest contribution made by the authors is the parametrization of the harm effect on the regret. To the best of my knowledge, no other works have proposed this.
The penalization $\lambda$, together with the threshold $\theta$ could allow practitioners to have more control on how much harm can be "risked" by taking an action.
This, I find to be a good original contribution.
Finally, the authors have provided the code used to generate the experiments. As such, the paper's results should be easily replicable.

**Weaknesses:**

From my understanding of the work, in a practical situation in which the value of $\lambda$ is high (i.e., the practitioners want to avoid harm at the cost of having a lower benefit) the initialization rounds of the algorithm would still cause some rounds to cause a high harm. As the authors also acknowledge in the conclusions, it would probably be possible to exploit independent samples collected during the execution of the algorithm to obtain a similar result in terms of convergence of the algorithm, while lowering the need for a long initialization phase.
As also noted by the authors, the expensive computation of $\eta$ could become impractical in a real use.
Finally, a minor observation, the graphs representing the average regret and average count of $p > \theta$ could benefit from representing also confidence intervals along with the mean.

**Questions:**

1) As the authors acknowledge in the conclusions, the balance between benefit and harm could be tackled through Pareto optimality. Turğay et. al. (2018) (https://doi.org/10.48550/arXiv.1803.04015) have proposed a method that exploits Pareto optimality. Although with some differences (Turğay et. al. allow for both conflicting and non-conflicting multiple objectives, but do not use penalizations and thresholds for harm), the two works share some commonalities. If the work of Turğay et. al. were to focus only on conflicting objectives, how would the authors compare the results obtained by the two works?

2) Can prior knowledge of $u_k$ be used to give a prior on reward and harm of the arms? Would it be possible to reduce the number of pulls in the initialization phase for ams which are known as "worse" with respect to $u_k$ while maintaining theorethical guarantees?

---

> ### Author Response · Authors · 2023-11-22
>
> Thank you for your detailed comments! We are happy that you find our parametrization of the harmful effect on the regret necessary and original. We address your questions below and will incorporate all feedback.
> * **The initialization rounds of the algorithm would still cause some rounds to cause high harm.**
> As we admit in Section 4 above Algorithm 1, initialization and exploration should be more carefully designed if the risk of causing harm is intolerable. In clinical trials, there are designs like 3+3, interval 3+3, and Bayesian optimal interval. These methods do not consider context but are widely used in practice to guarantee safety. We may use those as a reasonable initialization step. We currently do not assume prior information (knowledge from earlier phases of clinical trials, or data from clinical trials concerning a similar compound). But if some prior information is available, it can be used to make the algorithm safer, and possibly shorten the initialization stage. However, the purpose of this paper is to propose a framework considering harm. Tailoring the algorithm into an applicable clinical trial method will be interesting for future work.
> * **If the Pareto  MOMAB work of Turğay et. al. were to focus only on conflicting objectives, how would the authors compare the results obtained by the two works?**
> Please see the response to Reviewer GyGV. Our goal is to control the harm effect under the tolerable level $\theta$. So a treatment with both high harm and high reward may be in the Pareto optimal front, however unacceptable in the harm aspect.
> * **Confidence intervals along with the mean in the plots.**
> Thank you for the suggestion. We will add confidence bands in the plots.

---

### Official Review · Reviewer_BrFX · 2023-11-01

**Soundness:** 3 good
**Presentation:** 2 fair
**Contribution:** 2 fair
**Rating:** 5
**Confidence:** 3

**Summary:**

This paper addresses the problem of contextual-based online decision making with harmful effects. After parameterizing both the benefit and the harm effect, the authors propose an epsilon-Greedy algorithm that achieves regret of $\tilde{O}(\sqrt{T})$.

**Strengths:**

According to Table 1, this paper is the first work to consider feedback involving context features, action features, and underlying parameters in bandit problems. The authors have designed an algorithm that is provably achieving near-optimal regret in this setting.

**Weaknesses:**

1. The comparison to related work in this paper is not sufficiently clear. In the section comparing this work to previous MOMAB papers, the authors claim that "this paper is the first to consider MOMAB in a contextual setting." However, it seems that this paper revolves around a single-objective problem, as in equation (1) of the paper. This approach doesn't involve the optimization of Pareto regret, which as far as I know is the central topic in prior MOMAB works. Thus I think it is not so fair to make direct comparsion to previous MOMAB works.

2. The assumptions made in this paper is implicitly strong but not sufficiently clear: Theorem 1's results rely on the existence of a positive lower bound for $\lambda_{min}(\Sigma_1)$ and $\lambda_{min}(\Sigma_2)$. Implicitly, these bounds necessitate a diversity assumption concerning the distribution of context and action features. Such a strong assumption significantly simplifies the algorithm design of online decision-making. However, I was unable to locate this assumption clearly spelled out in Assumptions A1 to A3. It would be helpful if the authors could present such strong assumption in a more easily identifiable way.

**Questions:**

I have no further questions.

---

> ### Author Response · Authors · 2023-11-22
>
> Thank you for your insightful comments! We have incorporated all feedback.
> * **It is not so fair to make direct comparisons to previous MOMAB works.**
> Thank you for pointing this out. Although scalarization is a plausible approach to solve MOMAB, it is not the current focus of MOMAB (Pareto regret). Since originally we have 2 objectives, our work is related to MOMAB, and we would like to keep the MOMAB part in the literature review. But we have removed the claim and emphasized we do not work on the Pareto regret. We also think more deeply about why the Pareto regret may be not a good choice for us, see the response to Reviewer GyGV.
> * **The assumptions made in this paper are implicitly strong but not sufficiently clear.**
> Thank you for your careful thoughts. We have added assumption A4 for $\Sigma_1$ and $\Sigma_2$ to be positive definite.

---

### Official Review · Reviewer_GyGV · 2023-11-04

**Soundness:** 3 good
**Presentation:** 1 poor
**Contribution:** 3 good
**Rating:** 3
**Confidence:** 4

**Summary:**

The paper tackles the problem of multi-armed bandit(MAB) when the bandit's arms provide feedback as *reward* (positive effect) and *harm* (negative effect). The work uses a generalized linear model with contextual variables to model the rewards and harm and proposes a novel $\epsilon$-greedy algorithm to tackle the proposed MAB setup. The algorithm enjoys sublinear regret and is supplemented with extensive experimental evidence to support the claim.

**Strengths:**

The problem tackled in the paper is **very** significant as most of the applications witnessed in multi-armed bandits have an arm model where harm is witnessed but often neglected. This is an important direction for a sustainable future (for e.g. overconsumption is often associated with adverse effects)

The paper provides a novel method for modeling reward and harm in multi-armed bandits as well as a sublinear regret algorithm that can tackle the problem. The authors provide simulation evidence for the prowess of their method. They provide an explanation for the assumptions taken while proving the theorem.

**Weaknesses:**

I will split this answer into parts :

1) **Modelling choices**:  There is an opaqueness in the modeling choices at various places in the paper.

For e.g.
* The choice of $u_{i}$ for arms -- why are they scalar values, what is the physical interpretation, and why are they increasing across arms? Does that mean that arm $1$ and arm $K$ are best and worst or vice versa?

* choice of optimization problem taken in equation (1). Is there a motivation as to why equation (1) is chosen over other forms? Does the following work?
$$ \min \{\arg\max_k\{q_{k, t}^2 -\lambda(p_{k, t} -\theta)^2_+\}\} $$

* why is equation (5) linear in $u_k$ but equation (8) quadratic? Is it coming from some existing models?

These are some examples, I am not listing all of them

2) **Missing definitions**: Certain places seem to have missing definitions. For e.g.
* equation (4), (5), (7), (8) -- what are the functions p, q,g,h,$\zeta, \xi$? why are these equations the way they are?

3) **Missing lower bound**: There is no explanation or intuition as to why the upper bounds obtained in the paper could be tight. Would be good to have a quantification as to the suboptimality of the upper bounds in terms of the dimensional dependence.

4) **Algorithm ambiguity**: There are some points not covered when discussion of the algorithm happens:

* The algorithm is not really $\epsilon$-greedy. Maybe change the name or reference to $\epsilon_t$-greedy? Since $\epsilon$-greedy would give the impression of permanent forced exploration based on static $\epsilon$ which is not the case here.

* Why is forced exploration (through $\epsilon$-greedy methodology) required? Is it because of the lack of closed-form expression for confidence width?

* The choice of forced exploration parameter $\epsilon_t$ is taken to be $\propto \frac{\log T}{T}$ which is typically for the case of vanilla bandits as the confidence width there is also proportional to a similar function in $T$. Why is it the choice here when the confidence width is not discussed?

* There seems to be some mistake in the pseudocode line 3-5 (initialization phase).

* What is the complexity of each inner loop of the $\epsilon$-greedy algorithm?

5) **Limited experiments**: The experiments seem to be on a much smaller scale with no reasoning on issues taking it to a larger scale or to real-world datasets.

A suggestion would be to rename the algorithm to "
$\epsilon$-greedy" rather than "variable coefficient" (assuming they are the same)

It is completely possible that I might have missed some context while reading and I am open to changing my opinion on the issues listed

**Questions:**

1) I am a bit confused about the connection with Multiobjective MAB. Can the framework not tackle the setup of reward and harm? What are the exact challenges in extending the MOMAB framework to encompass the current problem?

2) There are a fair number of works on safety-constrained MABs, but I see limited mentions of them (E.g. some works focus on a safety budget). Are they very different from the current work and hence not mentioned?

---

> ### Author Response · Authors · 2023-11-22
> **Response to Reviewer GyGV**
>
> Thank you for your valuable and detailed comments! We are encouraged that you find our consideration about harm a very important question. We address your concerns below and will incorporate all feedback.
>
> **Modelling Choices**
> * **The choice of $u_i$ for arms:**
> $u_1\leq u_2\leq\cdots \leq u_K$ is just an assumption for simplicity. Otherwise one may re-order the arms to achieve this rank. It does not mean arm 1 to $K$ are best to worst, or vice versa. Since there is context information $X$, the best arm will be different for different values of $X$. So the order of $u_k$ does not imply the optimality of arms in any sense. The physical meaning of $u_k$ will depend on the problem at hand. For example, in clinical trials, $u_k$ may mean the dose level ($u_k$ mg/ml of medicine) at treatment $k$ (see 1st paragraph of section 3). This notion of $u_k$ is also used in [3, 20] as shown in table 1.
> * **Choice of optimization problem taken in equation (1):**
> Since the regret in multi-armed bandit literature is typically defined as the linear gap between the best arm and the chosen arm ($q^*_t-q_{A_t,t}$ in our notation), when we introduce another objective, we still define regret as the linear gap(equation 2). We do not see a motivation to modify the regret definition. Thus, it makes sense to define the optimization problem as a linear function instead of a quadratic function or other kind of functions. We also discussed in detail alternative choices of best arm in equation (3) and the paragraph below equation (3).
> * **Why is equation (5) linear but equation (8) quadratic? Is it coming from some existing models?**
> It stems from dose-repsonse models but we find it reasonable in general. We may assume the harm is increasing with the level of treatment (dose, for example), but benefit may not be monotone. So equation (5) for harm is chosen to be linear so that it is easy to make it monotonely increasing. Equation (8) is quadratic to allow increasing then decreasing behavior. The linear and quadratic functions are only examples of modeling choice, as explained below equation (8).
>
> **Missing lower bound**
> We do not have a lower bound for the regret ready. However, the bound of rate $\sqrt{T\log T}$ matches the state-of-art upper bound for generalized linear contextual bandit problems in literature.
>
> **Algorithm ambiguity**
> * $\epsilon_t$ greedy.
> Thank you for your suggestion. We have changed the algorithm name to $\epsilon_t$ greedy. We did not name "varying coefficient" as "$\epsilon_t$-greedy" as all the methods under comparison are under the $\epsilon_t$-greedy framework, but different modeling choices.
> * **Why is forced exploration (through $\epsilon$-greedy methodology) required?**
> As explained in Section 4 below Algorithm 1, since the scalarization makes the problem essentially nonlinear, an upper confidence bound based on the linear term(systematic component in GLM) cannot be constructed. Thompson sampling would be another approach, but we failed to get good results in some earlier simulation. Also, running MCMC for posterior with no closed form is expensive. It may be worthwhile to investigate how to finely tune MCMC and choose a good prior for this kind of models, but it's beyond the scope of this paper to propose a framework considering harm.
> * There seems to be some mistake in the pseudocode line 3-5 (initialization phase).
> We double checked and did not find a mistake. To make it more clear: We have $K$ arms, and we'd like to gather $m$ independent samples for each arm. For example, for time $t = 1,\cdots, m$, we pull arm 1, and for $t = m+1, \cdots, 2m$, we pull arm 2,...,and for  $t = (K-1)m+1, \cdots, mK$, we pull arm $K$. But overall, we just need to pull each arm $m$ times during $t=1,\cdots, mK$.
> * What is the complexity of each inner loop of the $\epsilon$-greedy algorithm?
> We need to fit GLM in each inner loop for $t-1$ data points at round $t$. Using the notation of the paper, the dimension of covariate for harm model is $2d_1$ ($\Phi(X_t)$ is $d_1$-dimensional) and the dimension of the covariate for reward model is $3d_2$, then the complexity for fitting 2 GLM is $O(t(d_1^3 + d_2^3))$.
>
> **Connection with Multiobjective MAB**
> Our scalarization formulation is from MOMAB, but with a specific focus on controlling harm, i.e., we penalize the harm probability when it exceeds the harm threshold. As for the popular Pareto formulation in MOMAB, it may not suit the needs of controlling harm very well. Consider a simple example, let $(p,q)$ denote a pair of mean harm $p$ and mean reward $q$, and there are totally two possible arms $(p,q) = (0.3, 0.7)$ and $(p,q) = (0.6, 0.8)$. Remember we prefer smaller $p$ and larger $q$. Then both arms $(0.3, 0.7)$ and $(0.6, 0.8)$ are in the Pareto optimal front, since they don't dominate each other. However, a harm of $p = 0.6$ may be unacceptable.
>
> **Safety-constrained MABs**
> We would appreciate it if you can provide some examples of safety budget bandit works.

---

> ### Author Response · Authors · 2023-11-22
> **Additional response to Reviewer GyGV**
>
> **Missing definitions**
> We have added more explanation in the paper.
> $p$ and $q$ are the means of harm and reward, respectively. $g$ and $h$ are the inverses of link functions for the GLM, as explained below in equations (5) and (8). Some examples of link functions are given in 3 paragraphs below equation (5).
> $\zeta$ and $\xi$ denote the systematic component of the GLM and their forms are defined in equations (5) and (8), respectively.
>
> **Limited experiments**
> We provide more experiment results in the supplement material. We did not carry out simulation for a larger time horizon $T$ since the algorithms are pretty stable at $T = 5,000$. Since there is no real experiment conducted according to the algorithm, we cannot provide real data analysis.

---

### Meta-Review · Area_Chair_qGAg · 2023-12-06

**Metareview:**

The paper considers contextual bandits with both rewards and harm constraints. The authors parameterize the rewards and harms with Generalized Linear Models and propose an $epsilon$-Greedy based algorithm. Theoretical analysis shows an $\tilde O(\sqrt{T})$ regret.
The reviewers recognize the significance of the problem and sound theoretical results. However, the reviewers share concerns about the unclear presentation, strong assumptions, unclear comparison with Multi-Objective MAB, and limited experiments. The rebuttal was helpful in clarifying some confusion but was not sufficient to resolve all concerns; specifically, the strong assumption on eigenvalues should be clearly discussed and justified. Considering the significance of the problem, the authors are encouraged to revise the paper following reviewers' suggestions in future submissions.

**Justification For Why Not Higher Score:**

Three out of four reviewers recommend rejection. The AC agrees with this decision. The reviewers share concerns about the unclear presentation, strong assumptions, unclear comparison with Multi-Objective MAB, and limited experiments; the rebuttal did not fully address the concerns.

**Justification For Why Not Lower Score:**

N/A

---

### Decision · Program_Chairs · 2024-01-16

Reject